# Effects of a Single Application of Scenar^TM^, a Low-Frequency Modulated Electric Current Therapy, for Pain Relief in Patients with Low Back and Neck Pain: A Randomized Single Blinded Trial

**DOI:** 10.3390/jcm10235570

**Published:** 2021-11-26

**Authors:** Mireille Michel-Cherqui, Avit Guirimand, Barbara Szekely, Titouan Kennel, Marc Fischler, Morgan Le Guen

**Affiliations:** 1Department of Anesthesiology and Pain Management, Hôpital Foch, France and Université Versailles-Saint-Quentin-en-Yvelines, 78000 Versailles, France; m.michel-cherqui@hopital-foch.com (M.M.-C.); b.szekely@hopital-foch.com (B.S.); m.leguen@hopital-foch.com (M.L.G.); 2Department of Anesthesiology, Hôpital Marie Lannelongue, Groupe Hospitalier Paris-Saint Joseph, 92350 Plessis-Robinson, France; mail@avit.fr; 3Department of Research and Innovation, Hôpital Foch, 92150 Suresnes, France; t.kennel@hopital-foch.com

**Keywords:** electric current therapy, low back pain, neck pain

## Abstract

We aimed to demonstrate the antalgic effectiveness of Scenar^TM^ (Self-Controlled Electro Neuro Adaptative Regulation) in patients experiencing low back and neck pain. Sixty patients were included and equally assigned by randomization to a Scenar-On group and to a Scenar-Off group (sham group). All patients received a 20 min application of Scenar^TM^ on the area where they experienced pain. The pain at rest and during movement and the sensation of stiffness were assessed using a numeric rating scale at baseline, immediately after the session and 24 h after the session. The patients’ characteristics at entry were similar between groups. The pain at rest decreased after the session in both groups (from 8 (4) to 5.0 (3) in the Scenar-Off group, *p* = 0.0001, and from 7 (3) to 4 (4) in the Scenar-On group, *p* < 0.0001). The difference was not statistically significant for the groups (*p* = 0.22). Similar results were observed during movement, but the sensation of stiffness was not modified. Such beneficial results did not last until the next day. No undesirable major effects were noticed. Our study does not support the fact that one Scenar^TM^ session improves low back and neck pain better than a sham session.

## 1. Introduction

The treatment of chronic pain is a major public health issue [1]. The relatively poor efficacy of medical treatments is even more concerning given the frequency of this pathology and its consequences on quality of life [2]. Consequently, this has led to the development of nonmedicinal and noninvasive treatments that are known to present less or no adverse effects. These treatments include low-power electrical currents applied to the surface of the skin in a noninvasive manner (electrostimulation), which are used mainly in the functional rehabilitation of nervous system injuries and for other purposes such as the treatment of pain. Some of these devices have obtained marketing authorization in several countries and are available without medical prescriptions. We therefore often see competition between medical channels and commercial channels facilitated by direct sales using websites.

The Self-Controlled Electro Neuro Adaptive Regulation (Scenar^TM^) device was developed by Soviet space program teams in the 1970s for reducing pain. It has been authorized by many regulatory authorities including the European Community in 2006 and the FDA in 2010. This electrotherapy device, which has some features close to a Transcutaneous Electrical Nerve Stimulator (TENS), is applied to the skin on an area of pain. Despite the lack of high-level publications, Scenar^TM^ is supposed to work similar to TENS by activating the descending pain modulation pathway and the gate control system and by liberating endorphins and endocannabinoids. Furthermore, the regular sliding of Scenar^TM^ can cause redness and a sensation of “stickiness” of the instrument in areas of pain or in trigger areas. These areas, which seem to be the sites of cellulalgia, are the areas where Scenar^TM^ should be applied successively for short periods of time using an individually dosed mode. In this mode, Scenar^TM^, which incorporates a feedback system, can measure the patient’s individual response to the electrostimulation and accordingly modify the upcoming impulses. Scenar^TM^ also has a mode of operation that causes a contraction of visceral smooth muscles, striated muscles and vessel walls, which could be responsible for a local anti-inflammatory effect, improvement of tissue oxygenation and restoration of muscle functions. Power, frequency and mode of impulsion can be set to induce different clinical effects. For example, the frequency varies between 15 and 340 Hz and it allows for stimulation that is more or less deep, but the standard frequency is around 90 Hz.

Scenar^TM^ is available for purchase on many sites, especially in the United States (https://ritmscenarusa.com/ Accessed date: 21 July 2021), in Switzerland (https://scenar.ch/ Accessed date: 21 July 2021) and in the European Community (http://ritm-europe.eu/ Accessed date: 21 July 2021). This explains why this device, or devices similar to this technology, is used by doctors, physiotherapists, osteopaths and sports trainers and directly by patients. Although these sites are assessed in a considerable number of publications, there have been few studies carried out with an accurate methodology on its effectiveness [3,4,5,6,7].

Although low back pain and neck pain have a major impact on health and social systems [8], only two studies have focused on these types of spinal pain, with conflicting results [3,6]. While the manufacturer recommends repeated application of Scenar^TM^ to treat pain, the aim of the present study was to evaluate the efficacy of a single application of Scenar^TM^ to decrease pain in patients suffering from neck or low back pain, since it appeared difficult to propose 5 to 10 sham sessions of Scenar^TM^ to patients. To the best of our knowledge, this type of study has never been performed.

## 2. Materials and Methods

This single-center, two-arm, parallel-group, randomized controlled superiority study was approved on September 10, 2018 by the Ethical Committee France Est II under the number 18/588, and patient enrolment was previously registered at ClinicalTrials.gov (NCT03755817—principal investigator: Mireille Michel-Cherqui—date of registration: 28 November 2018). Patients were included in the study after they gave their informed written consent. The study took place in a nonprofit tertiary care hospital. The authors adhered to the relevant reporting elements of the CONSORT statement (Appendix A).

### 2.1. Study Population

Patients, both male and female, aged between 18 and 80 years referred to a pain clinic for common acute or chronic low back pain or neck pain were included in the study. Neck pain also included cervicobrachial neuralgia, and low back pain also included lumbar radiculalgia. The excluded patients had the following characteristics: pregnant or breast-feeding; pacemakers or defibrillators; skin conditions making the use of Scenar^TM^ impossible (wounds, recent scars and skin infections); and low back or neck pain related to inflammatory, traumatic, oncologic, infectious or neurologic pathologies. Patients were not included if they had already had a Scenar^TM^ session. 

### 2.2. Randomization

The procedure used to allocate patients to the Scenar-On group or to the Scenar-Off group, which was the sham group, was managed by the sponsor (Direction of the research unit). A random allocation sequence, with a 1:1 ratio and blocks of 10, was generated using the site “Randomization.org”. Each patient received a unique patient number and a randomization number (patient code). For each subsequent patient, the investigator connected to a dedicated website using a protected password just after inclusion in the study. This system allocated the patients into groups.

### 2.3. Procedure

Each session lasted 20 min. A video recording, provided as Appendix A, demonstrates how the session was performed.

In the treatment group, the session was conducted according to the manufacturer’s recommendations and protocol. The power level of the device was set according to the patient’s sensations for the patient to remain comfortable during the whole procedure. During the first part of the treatment, the frequency was set at 90 Hz and the therapist gently moved the device regularly and with a slight compression on the areas described as painful by the patient. During this part, specific signs such as redness or pallor of the skin, sensation of stickiness of the device or an exacerbated pain usually appeared. During the second part of the treatment, the device was applied for about 30 to 120 s successively on the area where specific signs had been individualized after the sliding. In this mode, a beep signaled the completion of each treatment, and the device was positioned in an adjacent zone until all the areas had been treated.

In the control group, the procedure was identical, but the device stayed in the off position. 

Prior to commencement of the session, all participants were advised that they might not feel the electrotherapy during the trial, therefore minimizing any perception of a treatment occurring or not. Thus, in this single blinded study, patients were unaware if they had received sliding and successive application of the device plus electrotherapy (Scenar-On group), or sliding and successive application of the device alone (Scenar-Off group).

The full trial protocol can be obtained from the corresponding author upon request.

### 2.4. Data Collection

Clinical history was obtained from all individual participants included in the study and included demographic data, duration of pain, drugs and other treatments. Anxiety was evaluated using an 11-point numeric rating scale (NRS) from 0 (no anxiety) to 10 (maximum imaginable anxiety).

The intensity of self-reported pain at rest and at movement was measured using an 11-point numeric rating scale (NRS) from 0 (no pain) to 10 (maximum imaginable pain). The evaluation of stiffness of the neck or low back was assessed using an 11-point numeric rating scale (NRS) from 0 (no stiffness) to 10 (maximum imaginable stiffness). 

All evaluation data were collected by a professional unaware of the treatment allocation. These evaluations were obtained before the session, just after the session during a brief face-to-face interview and through a telephone call 24 h after the session. The patients were questioned about possible adverse effects just after the session and 24 h after the session. 

At the end of the study, the person in charge of the phone call was able to differentiation between the randomization, and a real session of electro-neuro-stimulation was proposed to the sham group. 

### 2.5. Outcomes

The main outcome of this study was pain intensity at rest just after a Scenar^TM^ session. The secondary outcomes included pain intensity at rest 24 h after the session, pain intensity during movement and sensation of stiffness just after the session and 24 h after, and adverse effects.

### 2.6. Statistical Analysis

#### 2.6.1. Sample Size Calculation

The analysis of our database allowed us to consider that the average pain score of patients suffering from back pain is 6 ± 2 (SD). Given the poor knowledge of the evolution of pain after Scenar^TM^ application, the number of patients to be included was calculated at 30 per group, taking into consideration an average pain improvement of 33% (20 on a 100 mm Visual Analog Scale (VAS) with a standard deviation of 2), a bilateral risk α of 0.05 and a power (1–β) of 0.8.

#### 2.6.2. Statistical Methods

All statistical analyses were performed on an intention-to-treat basis.

Categorical variables are presented as numbers (proportion) and compared between groups using a chi-square test or a Fisher’s exact test as appropriate. The distribution of continuous variables was assessed using the Shapiro–Wilk test. The normally distributed variables were summarized as mean ± standard deviation and compared between groups using the Student *t*-test, while the other variables were presented as median (interquartile range) and compared between groups using the Mann–Whitney test.

Repeated-measure mixed-model testing groups were performed to analyze pain parameters and changes of sensation of stiffness over time, including data obtained before the procedure, just after the procedure and the day after the procedure. A post hoc analysis was performed in case of a statistical significance.

All tests were two-sided. *p* values of less than 0.05 were considered significant. 

The statistics were generated using SAS 9.4 software (SAS Campus Drive Cary, NC, USA) and R software (version 3.1, R Foundation for Statistical Computing, Vienna, Austria) and using the application GMRC Shiny Stat (Strasbourg, France, 2017).

## 3. Results

### 3.1. Patients

The study took place from November 2018 to October 2020 in a tertiary private nonprofit hospital. A total of 60 patients were included in this study, 30 in the Scenar-On group and 30 in the Scenar-Off group, and were evaluated before the Scenar^TM^ application, just after the Scenar^TM^ application and the day after Scenar^TM^ application. All included patients received the treatment and were analyzed.

### 3.2. Patients’ Characteristics at Entry

The patients’ characteristics at baseline were similar between the groups, in particular the age in the Scenar-Off group (48 ± 14 years) and in the Scenar-On group (51 ± 3 years; *p* = 0.41) and the sex (76.7% of females in both groups). The cervical and low back pain locations were almost equally present in both groups. The preoperative use of painkillers was similar between the groups at baseline in this study (Table 1). The pain scores did not differ significantly between the groups at baseline (Table 1; *p* = 0.51 for the comparison of pain at rest and *p* = 0.96 for pain during movement).

### 3.3. Effects of the Scenar^TM^ Sessions

Pain at rest, the main outcome, decreased after the sessions in both groups (from 8 (4) to 5 (3) in the Scenar-Off group, *p* = 0.0001 and from 7 (3) to 4 (4) in the Scenar-On group, *p* < 0.0001). The difference of pain at rest was −2 (−2) in both groups (*p* = 0.22).

Similar results were observed during movement with a decrease in pain scores in both groups (from 8 (2) to 6 (4) in the Scenar-Off group, *p* = 0.0007 and from 7 (2) to 5 (4) in the Scenar-On group, *p* < 0.0001). The difference of pain at rest was −1 (−2) in the Scenar-Off group and −2 (−3.25) in the Scenar-On group (*p* = 0.12).

The sensation of stiffness was not modified (Figure 1). 

Group–time interactions were not statistically significant regardless of the variable concerned: pain at rest or during movement, or the sensation of stiffness. Interestingly, the group effect was never statistically significant, as opposed to the time effect, which was significant when considering pain at rest or during movement (Table 2).

### 3.4. Adverse Effects or Events

Just after the session, three patients in each group reported minor adverse effects: two cases of headache and one case of increase in back stiffness in the Scenar-On group, one case of fatigue, one case of increase in pain and one case of increase in back stiffness in the Scenar-Off group. Twenty-four hours after the session, five patients in the Scenar-On group reported minor adverse effects such as insomnia (one case), abdominal pain (one case), headache (two cases) and increase in pain (one case). Three patients in the Scenar-Off group reported minor adverse effects such as fatigue (one case), increase in pain (one case) and increase in back stiffness (one case). 

No patient reported serious adverse effects.

### 3.5. Follow-Up

At the end of the study, the patients were informed of their group of treatment. Nineteen patients among the thirty patients of the Scenar-Off group asked for a real session of electro-neuro-stimulation, and fourteen patients of the Scenar-on group asked for more treatment.

## 4. Discussion

There was a decrease in pain in patients involved in the study after one session of Scenar^TM^ treatment, but this beneficial effect was observed in both groups and did not persist to the following day. Consequently, our study does not support the fact that one “real” Scenar^TM^ session is superior to sham treatment in improving low back or neck pain. 

To our knowledge, our study is the first to compare a single session of Scenar^TM^ to a sham session. Other studies compared the effect of a similar but different device but did not compare the results to a sham group or evaluate the effect of multiple sessions. Among them, three randomized studies were conducted on neck pain. Our results are close to a preliminary study where Shabrun et al. compared a single session of interactive neurostimulation therapy to a sham treatment. They also observed improvements in pain intensity and neck disability in both the treatment and sham groups [5]. Other studies and reports are more in favor of a positive effect of this therapy. In a very small trial, participants who received a therapy close to Scenar^TM^ experienced greater reductions in the intensity of neck pain and disability and increased function and overall quality of life compared to participants receiving either TENS therapy or a sham treatment over 6 weeks [3]. The Scenar^TM^ therapy also proved to be superior to the TENS therapy in reducing pain and disability for whiplash injury after 6 weeks of treatment but was not compared to a sham group [6]. Interestingly, the after-sales survey of 481 people who bought and used an ENAR (a device similar to Scenar^TM^) reported a reduction in pain in 70% of cases and a functional improvement in 62% of cases [9]. 

Our results are especially marked by the similarity between the acute effects obtained by using the device in operation or not. This may be related to one of the possible mechanisms of the session, which is a “massage-like effect” due to the regular sliding of the device on a painful area. All types of manual therapy have been shown to elicit a neurophysiological response that is associated with the descending pain modulation circuit and the gate control system. Neurophysiological responses vary according to the type of therapy. The roles of endorphins, oxytocin and endocannabinoids have been advocated to explain the decrease in pain frequently described by patients [10,11]. This “massage-like effect” could be responsible for the decrease in pain noticed in the two groups and the absence of the specificity of one session. A powerful placebo effect could also be responsible for this effect, as we know that this kind of effect can vary between 30 and 50% according to the study in [12]. This effect could have been reinforced by the positive interaction between the patient and the nurse in charge of the treatment as all the nurses avoided negatively loaded suggestions and used routinely positive communication [13]. Finally, only minor side effects were observed in both groups.

### Strengths and Weaknesses 

The generalizability of our trial findings should not be a problem given the simplicity of the procedure, and our results could be used without difficulty by other teams while recalling that the manufacturer recommends repeated sessions and not a single session. However, some limitations should be discussed.

Although patients were warned before the application of Scenar^TM^ that they would not sense anything, clearly those who felt electric stimulation probably understood that they were in the Scenar-On group; this limits the blindness of the method but is hard, if not impossible, to avoid.

Our group of patients included those with low back pain and those with neck pain, with or without irradiation, but their numbers are too small for subgroup analysis. Furthermore, our population was not homogenous, and some patients experienced myofascial pain only, others presented a neuropathic component, and some of them were very anxious and tense. One could hypothesize that the different mechanisms that could explain the effectiveness of Scenar^TM^ (the electric stimulation, the sliding of the device and the empathic contact of the therapists) could have a different effect on the patients depending on the etiology of their pain. It is possible, but far from certain, that a more homogenous population could have helped us obtain clearer results.

Our methodology was that the first period of the session, with its “massage-like effect” (see Appendix A), was common to both groups and that the comparison concerns the second period during which the device was on or off. To clarify the respective effectiveness of these periods, only the second period could have been retained for this study, but this would not have corresponded to the description of a Scenar^TM^ session as proposed by the manufacturer.

Finally, the inclusion period was long, but this was explained by an interruption due to measures taken during the COVID-19 pandemic.

## 5. Conclusions

Our study, using a mixed model for repeated measures (before application, just after application and the day after application) to analyze the analgesic effect of one Scenar^TM^ session, does not support the fact that a single Scenar^TM^ session improves low back or neck pain. However, since repeated use of Scenar^TM^ is recommended by the manufacturer, new studies are needed to better define the effect of this device.

## Figures and Tables

**Figure 1 jcm-10-05570-f001:**
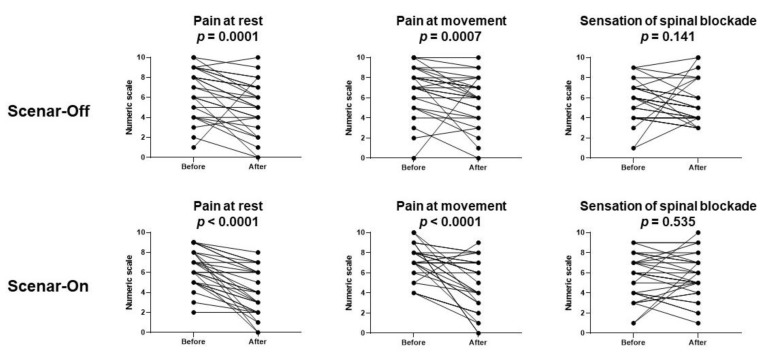
Evolution of pain scores and sensation of stiffness before and after the procedure. Scenar-Off: sham group with the Scenar^TM^ device turned off; Scenar-On: treatment group with the Scenar^TM^ device turned on. The data are presented as individual values.

**Table 1 jcm-10-05570-t001:** Patients’ characteristics at baseline.

	Scenar-Off Group (*n* = 30)	Scenar-On Group (*n* = 30)	*p*
Age (years)	48 ± 14	51 ± 13	0.41
Sex female	23 (76.7%)	23 (76.7%)	1
Body Mass Index	26.5 (6.2)	25.0 (5.0)	0.13
Pain			
More than 3 months of pain *	16 (53.3%)	12 (40.0%)	0.48
Location			0.60
Lumbar pain	15 (50.0%)	18 (60.0%)	
Cervical pain	15 (50.0%)	12 (40.0%)	
Lumbar pain irradiation			0.73
No irradiation	7 (23.3%)	7 (23.3%)	
Irradiation	8 (26.7%)	11 (36.7%)	
Cervical pain irradiation			1
No irradiation	3 (10.0%)	2 (6.7%)	
Irradiation	12 (40.0%)	10 (33.3%)	
Concomitant analgesics			
WHO analgesic level 1 *	18 (60.0%)	13 (43.3%)	0.30
WHO analgesic level 2 *	14 (46.7%)	10 (33.3%)	0.43
WHO analgesic level 3 *	1 (3.3%)	1 (3.33%)	1
Benzodiazepine	11 (36.7%)	6 (20.0%)	0.25
Antidepressant	11 (36.7%)	7 (23.3%)	0.40
Antiepileptic	7 (23.3%)	3 (10.0%)	0.30
Lidocaine 5% medicated plaster	3 (10.0%)	0 (0.0%)	0.24
NSAIDS	10 (33.3%)	5 (16.7%)	0.23
Other	1 (3.3%)	0 (0%)	1
Complementary therapies			
Transcutaneous electrical nerve stimulation	8 (26.7%)	5 (16.7%)	0.53
Auriculotherapy	2 (6.7%)	2 (6.7%)	1
Acupuncture	0 (0%)	2 (6.7%)	0.49
Physiotherapy	14 (46.7%)	17 (56.7%)	0.61
Psychocorporal techniques (relaxation, hypnosis and mindfulness)	7 (23.3%)	6 (20.0%)	1
Pain **			
At rest	8 (4)	7 (3)	0.51
During movement	8 (2)	7 (2)	0.96
Anxiety ***	5 (6)	4 (5)	0.15

*: World Health Organization (WHO) classification **: evaluated using an 11-point numeric rating scale (NRS) from 0 (no pain) to 10 (maximum imaginable pain) ***: evaluated using an 11-point numeric rating scale (NRS) from 0 (no anxiety) to 10 (maximum imaginable anxiety). The results are presented as mean ± standard deviation or median (interquartile range) depending on the normality of the distribution of continuous variables and numbers (percentage) for categorical variables.

**Table 2 jcm-10-05570-t002:** Effects of Scenar^TM^ sessions.

	Scenar-Off Group (*n* = 30)	Scenar-On Group(*n* = 30)	Mixed Model		
			Group Effect	Time Effect	Group-Time Effect
Pain at rest *			F = 2.527 *p* = 0.12	F = 36.445 *p* < 0.0001	F = 1.910 *p* = 0.15
Basal	8 (4)	7 (3)			
Postprocedure	5 (3)	4 (4)			
Day 1	6 (4)	4 (4)			
Pain at movement *			F = 1.742 *p* = 0.19	F = 17.981 *p* < 0.0001	F = 2.613 *p* = 0.08
Basal	8 (2)	7 (2)			
Postprocedure	6 (4)	5 (4)			
Day 1	7 (4)	6 (4)			
Stiffness **			F = 0.437 *p* = 0.51	F = 2.860 *p* = 0.44	F = 0.711 *p* = 0.49
Basal	6 (3)	6 (3)			
Postprocedure	5 (3)	6 (3)			
Day 1	5 (3)	5 (2)			

The results are presented as median (interquartile range). *: evaluated using an 11-point numeric rating scale (NRS) from 0 (no pain) to 10 (maximum imaginable pain). **: evaluated using an 11-point numeric rating scale (NRS) from 0 (no stiffness) to 10 (maximum imaginable stiffness). Day 1: the day after the procedure.

## Data Availability

The data are available on the Dryad website open-access repository (https://doi.org/10.5061/dryad.63xsj3v2s Accessed date: 25 August 2021).

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
