# Peer review of "Effects of a Single Application of ScenarTM, a Low-Frequency Modulated Electric Current Therapy, for Pain Relief in Patients with Low Back and Neck Pain: A Randomized Single Blinded Trial"

_jcm, 2021, doi:10.3390/jcm10235570_

Round 1
Reviewer 1 Report
This article is a re-submission about scenar device evaluated in patients with back or neck pain. The overall study is well describe in the manuscript.
One of the modifications is the aim of the study "to evaluate the efficacy of a single application of 69 Scenar to decrease pain in patients suffering from neck or low back pain since it appeared 70 difficult to propose to a patient 5 to 10 sham sessions of Scenar". But why is it hard to perform multiple sham sessions ? The main result of this study is negative but the manufacturer of the product could say that the protocol is incomplete, so noresults can be concluded to the study.
As previously mentionned, patients could know if they have the scenar on or off because they do not feel it (and it was probably mentionned in the information letter that the device could be on off-position). Probably that 19 patients have found that the device was not working (they ask for a "real" session). No patients in the scenar-on group ask it ? (Have to be noticed, even if the result is 0/30). This sentence indicate that this is not a real placebo research (who implies that patients sould not know the randomization)
It is not common to evaluate anxiety only using a VAS scale. No other data on comorbidities, especially psychological comorbidities, are available.
Even if there is an interest of a negative study on this product (because the manufacturer claim wonderful effects ont his website), and considering that the manuscript is globally well written, the main problem of a single-application design without clear justification, and a probably loss of placebo during the study induce that no definitive conclusion can be drawn
Author Response
Response to Reviewer 1
This article is a re-submission scenar device evaluated in patients with back or neck pain. The overall study is well describe in the manuscript.
Comment 1: One of the modifications is the aim of the study "to evaluate the efficacy of a single application of 69 Scenar to decrease pain in patients suffering from neck or low back pain since it appeared 70 difficult to propose to a patient 5 to 10 sham sessions of Scenar". But why is it hard to perform multiple sham sessions ? The main result of this study is negative but the manufacturer of the product could say that the protocol is incomplete, so no results can be concluded to the study.
Response to Comment 1:
- About the difficulty to propose several sham sessions of Scenar: We found it difficult to treat a patient only with Scenar for a long period of time. A protocol is being finalized with an active group and a sham group with an application every week for 5 weeks. Patients will be offered an alternative analgesic treatment, the dosage of which will be the primary endpoint.
- About the possible point of view of the manufacturer: Our title is not ambiguous: “Effect of a single application ….”. Furthermore, at the Reviewer's request, we have added the following sentence at the end of the introduction: “While the manufacturer recommends repeated application of ScenarTM to treat pain, the aim of the present study was to evaluate the efficacy of a single application of ScenarTM to decrease pain in patients suffering from neck or low back pain since it appeared difficult to propose to a patient 5 to 10 sham sessions of ScenarTM. This type of study has never been performed to the best of our knowledge.” Our initial text mentioned also in the Conclusion section: “… However, since repeated use of the Scenar is recommended by the manufacturer, new studies are needed to better define the effect of this device.” The manufacturer of the product could say that our protocol does not correspond to its recommendations, but our results are what they are and we are fully transparent.
Comment 2: As previously mentioned, patients could know if they have the scenar on or off because they do not feel it (and it was probably mentioned in the information letter that the device could be on off-position). Probably that 19 patients have found that the device was not working (they ask for a "real" session). No patients in the scenar-on group ask it ? (Have to be noticed, even if the result is 0/30). This sentence indicate that this is not a real placebo research (who implies that patients should not know the randomization).
Response to Comment 2: We agree with this remark, our sentence is confusing. Whatever the group, Scenar-On group or Scenar-Off group, each patient was informed after the end of the study of the group to which they had been assigned and were asked whether they wished to continue Scenar treatment in the Scenar-On group or start Scenar treatment in the Scenar-Off group. This has been corrected in the revised text: “At the end of the study, the patients were informed of their group of treatment, 19 patients among the 30 patients of the Scenar-Off group asked for a real session of electro-neuro stimulation and 14 patients of the Scenar-on group asked to pursue the treatment.”
The Reviewer returns to the reality of the placebo session. Patients were informed before the study that the device could be in the off or in the on position, but they were informed at the same time that they would feel the sliding of the device, and perhaps some other sensations. This already appears in the procedure paragraph: “Prior to commencement of the session, all participants were advised that they might not feel the electrotherapy during the trial, thereby minimizing any perception of a treatment occurring or not.”
Comment 3: It is not common to evaluate anxiety only using a VAS scale. No other data on comorbidities, especially psychological comorbidities, are available. YES
Response to Comment 3: We agree with that VAS scale is not usual when evaluating anxiety. However, this simple criterion has been shown to be useful. See the paper from Kindler et al which states that “VAS is a useful and valid method for measuring preoperative anxiety and compares well with the state anxiety score of the STAI” (The visual analog scale allows effective measurement of preoperative anxiety and detection of patients' anesthetic concerns. Anesth Analg. 2000 Mar;90(3):706-12). Thus this simple evaluation shows a similar level of anxiety in both groups. This seems sufficient to us as long as we did not use the level of anxiety as a factor in the analysis of the effectiveness of Scenar. We hope the Reviewer will share our opinion.
Comment 4: Even if there is an interest of a negative study on this product (because the manufacturer claim wonderful effects on his website), and considering that the manuscript is globally well written, the main problem of a single-application design without clear justification, and a probably loss of placebo during the study induce that no definitive conclusion can be drawn.
Response to Comment 4:
- Regarding the problem of the single application: Our Conclusion leaves no room for ambiguity: “Our study, using a mixed model for repeated measures (before application, just after and the day after) to analyze the analgesic effect of one ScenarTM session, does not support the fact that it improves low back or neck pain. However, since repeated use of the ScenarTM is recommended by the manufacturer, new studies are needed to better define the effect of this device.”
- Regarding the placebo: This is clearly a problem but what is the solution? Getting around the problem by comparing the Scenar to another device? We did not make that choice and prefer to use a sham session. This point has been added in the strength and weakness paragraph: “It is very difficult to determine an optimal comparator in a study evaluating an electrical device with a choice between the comparison of Scenar with another device or the use of a sham treatment. We decided to use the latter with the risk that we discussed of a “massage-like effect”.
Otherwise, the text has been read again by a native English copy editor.
Reviewer 2 Report
Dear editor,
This kind of research is of high value to get insight of instruments like this which are on the market. However, I’ve methodological and statistical concerns related to this research.
Methodological:
It is of importance that the authors describe their paper based on the CONSORT Statement (http://www.consort-statement.org/); the checklist of information to include when reporting a randomised trial. So please revise the whole manuscript related to the items in this checklist and describe all revisions related to each CONSORT-item in your rebuttal. Thereby describe in the first paragraph of your Methods that the CONSORT Statement was used.
Statistical:
It is not clear why the Wilcoxon test is used. Are the outcome parameters seen as ordinal outcomes or are the used outcome parameters not normal distributed?
Related to Table 1 you give p-values for the pain-scores. However it is not described in the methods section which statistical test(s) were used to test this.
It is better to compare both groups by statistical testing for all mentioned items in Table 1. So for numerical data the chi-square test, or the Fisher Exact Test when N<5 in one cel. For normal distributed continues data the independent t-test. For ordinal data and non- normal distributed continues data the Mann-Withney U test.
Author Response
Response to Reviewer 2
Methodological comment: It is of importance that the authors describe their paper based on the CONSORT Statement (http://www.consort-statement.org/); the checklist of information to include when reporting a randomised trial. So please revise the whole manuscript related to the items in this checklist and describe all revisions related to each CONSORT-item in your rebuttal. Thereby describe in the first paragraph of your Methods that the CONSORT Statement was used.
Response to the Methodological comment: We have added as a Supplementary file the CONSORT checklist of information and verify that we have followed it. Furthermore, we have added the following sentence at the end of the first paragraph of in the Materials and Methods section: “The authors adhered to the relevant reporting elements of the CONSORT statement (Supplementary file 1).”
Statistical comment :
It is not clear why the Wilcoxon test is used. Are the outcome parameters seen as ordinal outcomes or are the used outcome parameters not normal distributed?
Related to Table 1 you give p-values for the pain-scores. However it is not described in the methods section which statistical test(s) were used to test this.
It is better to compare both groups by statistical testing for all mentioned items in Table 1. So for numerical data the chi-square test, or the Fisher Exact Test when N<5 in one cel. For normal distributed continues data the independent t-test. For ordinal data and non- normal distributed continues data the Mann-Whitney U test.
Response to the Statistical comment: We thank the Reviewer for pointing out errors in our analysis. This led us to rewrite the Statistical analysis section as follows: “All statistical analyses were done on an intention to treat basis. Categorical variables are presented as number (proportion) and compared between groups using Chi-square test or Fisher’s exact test as appropriate. Distribution of continuous variables was assessed using the Shapiro-Wilk test. Normally distributed variables were summarized as mean ± standard deviation and compared between groups using the Student t-test while the other variables were presented as median [interquartile range] and compared between groups using the Mann-Whitney test. Repeated-measures mixed-models testing groups were performed to analyze pain parameters and sensation of stiffness changes over time including data obtained before the procedure, just after it and the day after. Post-hoc analysis was performed in case of statistical significance. All tests were two-sided. P values of less than 0.05 were considered significant. The statistics were generated using SAS 9.4 software.”
Accordingly, we have corrected the Results and Table 1. We have also compared patients' characteristics at baseline as requested by the Reviewer.
Round 2
Reviewer 1 Report
The authors have made significant changes in their manuscript, especially on statistical data. This is surprizing that two different statistical softwares were used, suggesting that statistical analysis was made by two different person in the initial and review manuscript.
The authors clarify one sentense who suggested in the initial manuscript that patients know that they have been in placebo arm. Not all the problem with placebo are resolved, but this sentense is more understandable.
In conclusion, this study, using a design of a unique seance of the device, and only simple evaluations (such as anxiety evaluate in only one question) is a "preliminary study" (those words are used by the authors in their response, but not in the manuscript). As a preliminary study, the manuscript has been improved with the review.
Author Response
- The authors have made significant changes in their manuscript, especially on statistical data. This is surprising that two different statistical softwares were used, suggesting that statistical analysis was made by two different person in the initial and review manuscript.
Response > We did modify the statistical analysis as requested by the Reviewer. This was the comparison between the Table1 data. These modifications were done with another software program that was used to do basic statistical tests. These modifications do not change the results of the Scenar effect analysis.
- The authors clarify one sentence who suggested in the initial manuscript that patients know that they have been in placebo arm. Not all the problem with placebo are resolved, but this sentence is more understandable.
Response > We agree with this comment.
- In conclusion, this study, using a design of a unique séance of the device, and only simple evaluations (such as anxiety evaluate in only one question) is a "preliminary study" (those words are used by the authors in their response, but not in the manuscript). As a preliminary study, the manuscript has been improved with the review.
Response > We agree with this comment.
Reviewer 2 Report
The authors revised their paper well. I only miss information about generalisability (external validity, applicability) of the trial findings (item 21 of the CONSORT statement) in the discussion.
So what are your advices for clinicians (applicability) of the Scenar and how close are the outcomes of this study to the clinical situation (external validity). (The internal validity is of good quality in your study, because you performed a RCT of good quality)
Author Response
- The authors revised their paper well. I only miss information about generalisability (external validity, applicability) of the trial findings (item 21 of the CONSORT statement) in the discussion.
Response > We wrote in our manuscript: "Our results could be used without difficulty by other teams.” This corresponded, for us, to answer the item pointed out by the Reviewer. To be clearer, we have modified the sentence as follows: "Generalizability of our trial findings should not be a problem given the simplicity of the procedure and our results could be used without difficulty by other teams while recalling that the manufacturer recommends repeated sessions and not a single session."
- So what are your advices for clinicians (applicability) of the Scenar and how close are the outcomes of this study to the clinical situation (external validity). (The internal validity is of good quality in your study, because you performed a RCT of good quality)
Response > The response to this comment is included in the revision made in response to the previous question.
This manuscript is a resubmission of an earlier submission. The following is a list of the peer review reports and author responses from that submission.
Round 1
Reviewer 1 Report
This study is a randomized single blinded trial to demonstrate the effect of a single application of Scenar for pain relief in patients with low back pain and neck pain. However, it has several problems in the study design and conclusion.
It was not clear why the authors investigated the effect of a single application of Scenar because several previous studies used different application methods.
The included patients not were appropriate to demonstrate the effect of a single application of Scenar because there were various causes of acute or chronic, neck or back pain.
The authors provided massage in addition to electric current therapy. However, massage was difficult to be controlled for uniform application and could be a bias of the results if the therapists were not blinded.
All outcome measures were subjective and it was not appropriate to conclude that the transient effect was possibly because of the massage without additional objective outcome measures.
Author Response
Comment 1. It was not clear why the authors investigated the effect of a single application of Scenar because several previous studies used different application methods.
Response to Comment 1. It seemed important to us to determine first the effectiveness of a single session of ScenarTM. We consider that our study is a preliminary one before considering the evaluation of a repetitive treatment (see the conclusion of the revised version: “Our study, using a mixed model for repeated measures (before application, just after and the day after) to analyze the analgesic effect of one ScenarTM session, does not support the fact that it improves low back or neck pain. However, since repeated use of the ScenarTM is recommended by the manufacturer, new studies are needed to better define the effect of this device.”
To best of our knowledge, no such study has ever been published. We have modified the text to respond to this comment: “The aim of the present study was to evaluate the efficacy of a single application of ScenarTM to decrease pain in patients suffering from neck or low back pain; such study has never been performed to the best of our knowledge.”
Comment 2. The included patients not were appropriate to demonstrate the effect of a single application of Scenar because there were various causes of acute or chronic, neck or back pain.
Response to Comment 2. We agree with this comment. The heterogeneity of the patients included in this study is indeed a limitation of this study which is indicated in the chapter devoted to its weaknesses: “Our group of patients included those with low back pain and those with neck pain, with or without irradiation, but their numbers are too small to perform a subgroup analysis. Furthermore, our population is quite heterogeneous as it included patients who presented myofascial pain but some of them presented also some criterion of neuropathic pain, and this had not been taken into account.”
Comment 3. The authors provided massage in addition to electric current therapy. However, massage was difficult to be controlled for uniform application and could be a bias of the results if the therapists were not blinded.
Response to Comment 3. Our text was ambiguous to say the least. Massage is not added to electric current therapy; the regular sliding of the device induces specific local signs and is responsible for a “massage like effect” .This is an integral part of the Scenar application. In fact, the investigators had an identical practice of the session whether the device was “on” or “off”. This is better explained in the revised version:
- in the Introduction section: “Furthermore, the regular sliding of the ScenarTM can cause redness and a sensation of “stickiness” of the instrument in areas of pain or in trigger areas. These areas, which seem to be the site of cellulalgia, are the area where the ScenarTM should be applied successively for short periods of time, using an individually dosed mode.”
- in the Procedure section: “During this part, specific signs as redness or pallor of the skin, sensation of stickiness of the device, or an exacerbated pain usually appeared.”
On the other hand, we mention a “massage-like effect” in the Discussion section: “Our results are especially marked by the similarity between the effect obtained in acute by the device whether it is in operation or not. This may be related to one of the possible mechanisms of the session which is a “massage-like effect” due to the regular sliding of the device on the painful area. All types of manual therapy have been shown to elicit a neurophysiological response that is associated with the descending pain modulation circuit and the gate control system. Neurophysiological response varies according to the type of therapy. The role of endorphin, oxytocin and endocannabinoid has been advocated to explain the decrease in pain frequently described by the patients [10,11]. This “massage –like effect” could be responsible for the decrease in pain noticed in the two groups and the absence of specificity of one single session.”
Comment 4. All outcome measures were subjective and it was not appropriate to conclude that the transient effect was possibly because of the massage without additional objective outcome measures.
Response to Comment 4. As mentioned above, the massage effect has been specified. In any case, pain assessment is subjective. However, the fact that the patients were blinded to the technique gives value to our results.
Reviewer 2 Report
This manuscript evaluate a local device, called Scenar, on neck and back pain.
The overall quality of the study is good, the design of the study is appropriate. However, I have some concern about various points :
- Description of the mechanism of the device is lacking in the introduction.
- The patients could know if they have the scenar-on or off because it induce a specific sensation on the skin.
- Problem with sample size (probably 20 on a 100 128 mm Visual Analog Scale (VAS) and not 2 !
There is some spelling mistakes on the manuscript, and some unusual terms (for example table 1 : characteristics at baseline more than entry; in the table 1 NSAIDS and not NIADS/ “their numbers are too small to do a subgroup analysis » “perform” more than “do” ?)
In the table 1, why anxiety is reported but not pain ?
Is pain not different in the two groups at baseline ?
Sentense “No patient reported serious adverse effects and 19 patients of the Scenar-Off group 197 asked for a real session” is surprizing. Is the information of the on/off group has been given to the participants during the research ?
I don’t understand the conclusion “Our study does not support the fact that one Scenar session improves low back or 264 neck pain but has a transient effect which is possibly because of the massage” because there is no significant differences just after the scenar application ? What is this transient effect ?
Before this review, I had no idea about this device. On their website, they claim wondeful results on pain but also on multiple other diseases witch are very surprizing. On their english website, on the palliative oncologic part : "The results of SCENAR-therapy application for detoxication, chronic pain relief, manifestations of fatigue, respiratory compromise, intracavitary effusions, anemic cardiopathy, and infectious complications as well as for fighting against tumorous ulcers". So a negative article on this device is probably in the public interest
Author Response
Comment 1. Description of the mechanism of the device is lacking in the introduction.
Response to Comment 1. We have added in the Introduction section some precisions on the supposed mechanisms of the device: “The Self-Controlled Energo-Neuro Adaptive Regulation device (ScenarTM) was developed in the 1970s by Soviet space program teams for reducing pain. It has been authorized by many regulatory authorities including the European Community in 2006 and the FDA in 2010. This electrotherapy device, which has some features close to a Transcutaneous Electrical Nerve Stimulator (TENS), is applied on the skin on an area of pain. Despite the lack of high-level publications, ScenarTM is supposed to work like TENS, by activating the descending pain modulation pathway and the gate control system and by liberating endorphins and endocannabinoids. Furthermore, the regular sliding of the ScenarTM can cause redness and a sensation of “stickiness” of the instrument in areas of pain or in trigger areas. These areas, which seem to be the site of cellulalgia, are the area where the ScenarTM should be applied successively for short periods of time, using an individually dosed mode. In this mode, the ScenarTM, which incorporates a feedback system, can measure the patient’s individual response to the electrostimulation and it accordingly modifies the upcoming impulses. The ScenarTM also has a mode of operation which causes a contraction of visceral smooth muscles, striated muscles, and vessel walls, which could be responsible for a local anti-inflammatory effect, improvement of tissue oxygenation and restoration of muscle functions. Power, frequency, and mode of impulsion can be set in to induce different clinical effects. For example, the frequency varies between 15 and 340 Hz, and it allows stimulation more or less deeply, but standard frequency is around 90 Hertz.”
Comment 2. The patients could know if they have the scenar-on or off because it induce a specific sensation on the skin.
Response to Comment 2. This is indeed a limitation of the method; it is added in the text in the section Strength and weakness: “Although patients were warned before the application of the ScenarTM that they could not sense anything, it is clear that those who felt electric stimulation probably understood that they were in the Scenar-On group; this limits the blindness of the method but is hardly if not impossible to avoid.”
Comment 3. - Problem with sample size (probably 20 on a 100 mm Visual Analog Scale (VAS) and not 2 !
Response to Comment 3. This error has been corrected.
Comment 4. There is some spelling mistakes on the manuscript, and some unusual terms (for example table 1 : characteristics at baseline more than entry; in the table 1 NSAIDS and not NIADS/ “their numbers are too small to do a subgroup analysis » “perform” more than “do” ?)
Response to Comment 4. These mistakes have been corrected.
Comment 5. In the table 1, why anxiety is reported but not pain ?
Response to Comment 5. Pain scores have been reported in Table 2 (mixed model analysis). We understand that pain scores at baseline have also their places in Table 1 (characteristics at baseline). They have been added.
Comment 6. Is pain not different in the two groups at baseline ?
Response to Comment 6. Our statisticians recommend that data from two groups should not be compared at baseline in randomized studies. Therefore, Table 1 presents the Patients' characteristics at baseline without comparing them. However, we understand the importance of this commentary; this has led us to modify the text: "Pain scores did not differ significantly between groups at baseline (Table 1; p=0.511 for the comparison of pain at rest and p=0.956 for pain at mobilization).”
Comment 7. Sentence “No patient reported serious adverse effects and 19 patients of the Scenar-Off group asked for a real session” is surprizing. Is the information of the on/off group has been given to the participants during the research ?
Response to Comment 7. We agree that this sentence is confusing. Patients were informed of their group at the end of the study. To avoid any misunderstanding, this sentence has been modified as follow: “At the end of the study, 19 patients among the 30 patients of the Scenar-Off group asked for a real session of electro-neuro stimulation.” This sentence has been cited in a subchapter entitled “3.5. Follow-up”.
Comment 8. I don’t understand the conclusion “Our study does not support the fact that one Scenar session improves low back or neck pain but has a transient effect which is possibly because of the massage” because there is no significant differences just after the scenar application ? What is this transient effect ?
Response to Comment 8. We agree that the conclusion was not clear, and we have modified it: “Our study, using a mixed model for repeated measures (before application, just after and the day after) to analyze the analgesic effect of one ScenarTM session, does not support the fact that it improves low back or neck pain. However, since repeated use of the ScenarTM is recommended by the manufacturer, new studies are needed to better define the effect of this device.”
Comment 9. Before this review, I had no idea about this device. On their website, they claim wonderful results on pain but also on multiple other diseases witch are very surprizing. On their english website, on the palliative oncologic part : "The results of SCENAR-therapy application for detoxication, chronic pain relief, manifestations of fatigue, respiratory compromise, intracavitary effusions, anemic cardiopathy, and infectious complications as well as for fighting against tumorous ulcers". So a negative article on this device is probably in the public interest.
Response to Comment 9. This comment is in line with our Introduction where it is written: “… These treatments include low power electrical currents applied to the surface of the skin in a non-invasive manner (electrostimulation), which are used mainly in functional rehabilitation of nervous system injuries and in other indications such as the treatment of pain. Some of these devices have obtained marketing authorization in several countries and are available without medical prescriptions. We therefore often see competition between medical channels and commercial channels facilitated by direct sales using websites.”
Round 2
Reviewer 1 Report
In the response to comment 1, the authors responded that “It seemed important to us to determine first the effectiveness of a single session of ScenarTM “. I think that those purpose did not provide any further information for the readers because the manufacturer’s recommendation and previous studies used the repeated application of ScenarTM.
In the response to comment 2, the authors responded that “The heterogeneity of the patients included in this study is indeed a limitation of this study which is indicated in the chapter devoted to its weaknesses.“ The authors should consider the etiology of low back and neck pain because the authors provide both electrical stimulation and massage-like effect by ScenarTM.
In the response to comment 3 and 4, the authors responded that massage was not added and the regular sliding of the device was “massage-like effect” during application of ScenarTM. The authors also responded that “massage –like effect could be responsible for the decrease in pain noticed in the two groups and the absence of specificity of one single session.” The authors need a detail study design to avoid the bias of massage-like effect.